# Cultural Competence of Obstetricians/Gynecologists and Midwives Providing Midwifery Care to Roma Women in Western Greece

**DOI:** 10.3390/healthcare13020190

**Published:** 2025-01-19

**Authors:** Chrysoula Chinoporou, Athina Diamanti, Eleni Asimaki, Christina Nanou, Pinelopi Varela, Victoria Vivilaki, Anna Deltsidou

**Affiliations:** Department of Midwifery, Faculty of Health and Caring Sciences, University of West Attica, 12243 Egaleo, Greece; aebmc21057@uniwa.gr (C.C.); easimaki@uniwa.gr (E.A.); nanouxv@uniwa.gr (C.N.); pvarela@uniwa.gr (P.V.); vvivilaki@uniwa.gr (V.V.); adeltsidou@uniwa.gr (A.D.)

**Keywords:** Roma, obstetricians/gynecologists, midwives, cultural competence

## Abstract

**Background:** Providing midwifery care to Roma women is a significant public health issue due to their status as a vulnerable population, often facing unique challenges and discrimination in accessing healthcare. Cultural competence refers to the ability of maternity providers to understand and incorporate cultural factors within the broader healthcare system. **Objective:** This study aimed to investigate the cultural competence of obstetricians/gynecologists and midwives working in Western Greece who provide midwifery care to Roma women. **Methods:** A cross-sectional quantitative study was conducted using a questionnaire from the Roma Women’s Empowerment and Fight against discrimination in Access to Health (REACH) project, which focuses on empowering Roma women and combating healthcare access discrimination. The questionnaire covered three areas: the cultural competence of maternity professionals, their knowledge of Roma women’s lifestyle, and participants’ demographics. The sample included 100 maternity professionals from hospitals and health centers in Western Greece. **Results:** Cultural competence was found to be moderate to high, with a mean score of 6.9 (SD = 2.2) for the ability to provide adequate care. In the past six months, 33% had provided care to 1–5 Roma women with communication issues, and 53% frequently faced challenges in service delivery. Common problems included Roma women not understanding the information provided (72.9%), and not having necessary documents (41.7%). Obstetricians/gynecologists had higher knowledge scores compared to midwives, and higher educational attainment correlated with better knowledge. Older age was associated with higher cultural competence (*p* = 0.048). **Conclusions:** Cultural competence was positively correlated with knowledge levels, with obstetricians/gynecologists exhibiting higher competence than midwives. Enhancing cultural competence among perinatal care providers is crucial to addressing health disparities faced by Roma women. The study’s cross-sectional design and reliance on self-reported data may limit the generalizability and introduce bias. Enhancing cultural competence through targeted training programs can help address healthcare disparities faced by Roma women.

## 1. Introduction

Cultural competence is increasingly recognized as a cornerstone of effective healthcare delivery, particularly in multicultural societies where patients present diverse ethnicities, languages, traditions, and health beliefs [1]. Defined as “the ability of individuals to establish effective interpersonal and working relationships that supersede cultural differences” [2], cultural competence enables healthcare providers to respect and integrate the cultural, social, and linguistic needs of patients. By addressing the multifaceted social determinants of health and fostering meaningful patient–provider relationships, cultural competence contributes to the reduction of health disparities and the improvement of patient satisfaction and equity in healthcare outcomes [2].

The components of cultural competence include cultural awareness, cultural knowledge, cultural skills, cultural encounters, and cultural desire [3]. These elements collectively promote the ability of healthcare providers to deliver personalized, patient-centered care in diverse settings. Evidence suggests that interventions aimed at enhancing cultural competence improve healthcare providers’ knowledge, attitudes, and skills, and positively influence patient satisfaction [3,4]. However, gaps remain in linking these outcomes to tangible improvements in patient health and adherence, highlighting the need for further research to optimize educational strategies and measure patient-centered outcomes comprehensively.

The Roma community, often referred to as Romani, is an ethnic minority with origins tracing back to the Indian subcontinent. Linguistic and genetic evidence suggests that they migrated to Europe over a millennium ago, during the early medieval period. Today, they are one of the largest and most dispersed ethnic minorities in Europe, with significant populations across various countries. The Roma people have a rich cultural heritage characterized by unique traditions, languages (such as Romani), and a nomadic history [5,6].

In Europe, the Roma community represents one of the largest ethnic minorities, yet they are often marginalized and face significant barriers to accessing healthcare. This marginalization is compounded by cultural differences that can lead to misunderstandings or mistrust between Roma patients and healthcare providers [2,3]. For Roma women, who often require specialized care during pregnancy and childbirth, these barriers can result in inadequate care and poor health outcomes [7].

In Greece, the Roma are an integral but marginalized part of the population. Estimates suggest that they comprise approximately 2–3% of the Greek population. While exact numbers vary due to a lack of consistent census data, it is estimated that there are about 200,000–300,000 Roma individuals in Greece [7,8]. As in many other European countries, Roma women in Greece frequently encounter challenges when seeking midwifery care. These challenges stem from a combination of socio-economic disadvantages, cultural practices that differ from the mainstream, and discrimination within the healthcare system [9]. The Roma community faces significant challenges, including profound health disparities driven by socioeconomic disadvantages, such as poverty, inadequate housing, and limited access to clean water, which contribute to higher rates of disease and poor maternal and child health outcomes. Health literacy is a critical barrier, with limited knowledge about healthcare rights and preventive services leading to underutilization and difficulty navigating healthcare systems. Cultural competence gaps among healthcare providers often result in misinterpretation of Roma cultural practices, reinforcing stereotypes and mistrust. Discrimination and stereotyping, both at an individual and institutional level, further marginalize Roma individuals, while structural barriers, such as the lack of Roma mediators and bureaucratic challenges, continue to restrict their access to equitable healthcare services [9]. The concept of cultural competence among healthcare providers—defined as the ability to understand, respect, and effectively interact with people from different cultures—is essential in bridging these gaps and ensuring that Roma women receive the care they need [4].

The importance of cultural competence in healthcare is underscored by the complex needs of Roma women, whose health practices and beliefs may not align with conventional medical practices. Without an adequate understanding of these cultural nuances, healthcare providers may inadvertently contribute to the disparities that Roma women face in accessing care [10]. Studies have shown that healthcare providers who are culturally competent are better equipped to engage with Roma patients in a manner that is respectful and responsive to their unique needs, thereby improving health outcomes [11].

This article explores the current state of cultural competence among obstetricians/gynecologists and midwives in Western Greece, focusing on how these healthcare providers interact with and care for Roma women. Understanding the cultural competence of these providers is vital for identifying areas where improvements can be made, ultimately leading to better healthcare experiences and outcomes for Roma women.

## 2. Methods

### 2.1. Study Design and Population

This is a cross-sectional quantitative study, conducted between June 2023 and October 2023. The sample consisted of 100 individuals. The participants were midwives and obstetricians/gynecologists who work in facilities (hospitals and health centers) in Western Greece. The sample for the study was collected through convenience sampling. This non-probability sampling approach selected individuals who were readily accessible and willing to participate, making it a practical choice for exploratory research. While convenience sampling facilitates data collection in specific contexts, it may limit the generalizability of findings due to potential selection bias.

Inclusion criteria were professionals actively involved in providing midwifery or obstetric care to Roma women within the region. Exclusion criteria included healthcare professionals without direct contact or experience working with Roma populations. This ensured that participants had relevant exposure to the context under investigation.

### 2.2. Data Collection Method

The data collection was carried out through an in-person approach at healthcare facilities, including hospitals and health centers, where midwives and obstetricians/gynecologists were employed. Researchers scheduled visits to these locations and approached eligible participants during breaks or at times pre-arranged with department heads to minimize disruption to clinical duties.

Participants were provided with a brief verbal explanation of the study’s purpose and relevance, emphasizing the importance of their input in understanding cultural competence in maternity care for Roma women. To ensure participants felt comfortable, researchers explained that responses would remain anonymous, and no identifiable information would be linked to the questionnaire data.

To facilitate convenience and ensure confidentiality, participants were given the option to complete the questionnaire on-site or take it to complete later in a private setting. A secure collection method was established, where completed questionnaires could be placed in sealed envelopes or collected in a locked drop box provided at each facility. The envelopes or drop boxes were retrieved by the research team at designated intervals.

Each participant received an approximate timeframe for questionnaire completion, ensuring they did not feel rushed. Researchers remained available at the collection sites to address any logistical concerns or clarify queries without influencing the participants’ responses.

### 2.3. Data Collection Tool

For the purposes of the study, a questionnaire was used which focuses on the empowerment of Roma women and the fight against discrimination in health access. Permission to use the questionnaire was obtained from the researchers who developed the questionnaire. The researchers in that study investigated the healthcare access issues and challenges faced by Roma women in Greece, focusing on sexual and reproductive health. Using mixed methods, they employed a questionnaire that was validated for healthcare and other professionals, as well as interviews and focus groups, and revealed themes like stereotypes, cultural communication barriers, and the need for systemic changes and Roma mediators [9].

The questionnaire consists of a total of 46 items and is divided into 3 sections. The first section includes questions related to the cultural competence of health professionals regarding Roma women and consists of 20 questions in total, 4 of which concern case studies of clinical incidents and their management by the respective health professional. The second section includes questions regarding the participants’ knowledge of the lifestyle and habits of Roma women and consists of 13 questions. The third section includes 13 questions for collecting demographic data. The questions are closed-ended and multiple-choice. The questions in the first and second sections use a variety of Likert scales to assess participants’ responses. These include a 10-point scale ranging from “Not at all” to “Extremely” for assessing capability in providing care, and a 5-point Likert scale ranging from “Not at all” to “Absolutely”, as well as “Strongly disagree” to “Strongly agree”, depending on the question type. Additionally, participants were asked about their perceptions and experiences using multiple-choice questions and categorical responses, such as frequency (e.g., “Always”, “Frequently”, and “Rarely”), and demographic questions to collect information like age, gender, education, and workplace.

### 2.4. Research Ethics and Required Actions for Licensing from Official Scientific and Administrative Bodies

During the conduct of the research, all necessary ethical and deontological parameters were adhered to. The management and processing of the personal data of the study participants were conducted in accordance with the provisions of the General Data Protection Regulation (GDPR, EU 2018/1725). In accordance with the Principle of Data Minimization, the data collected were limited to what was necessary for the purposes of the study. The questionnaires were anonymous and were distributed to obstetricians/gynecologists and midwives working in facilities (hospitals, health centers) in Western Greece who met the study criteria. Written consent was obtained from them for participation in the study after they were informed.

For the conduct of the study, approval was obtained from the Research Ethics Committee of the University of West Attica (protocol number: 61445/27-06-2023).

### 2.5. Statistical Analysis

The choice of statistical methods was guided by the need to accurately describe the data and ensure robust comparisons aligned with the study’s objectives. The Kolmogorov–Smirnov test was employed to assess the normality of continuous variables, as it is crucial to determine whether parametric or non-parametric tests should be applied, ensuring valid statistical inferences. For variables with normal distribution, means and standard deviations (SDs) provided a comprehensive summary of central tendency and dispersion, suitable for identifying patterns in the data. In cases where variables deviated from normality, medians and interquartile ranges (IQRs) offered a robust alternative that reduced sensitivity to outliers.

The selection of the Student’s *t*-test and Analysis of Variance (ANOVA) for parametric comparisons reflected the importance of leveraging their sensitivity to detect differences between groups when normality was met. Conversely, the Mann–Whitney and Kruskal–Wallis tests were preferred for non-parametric data to account for skewed distributions. The application of the Bonferroni correction for multiple comparisons demonstrated a commitment to reducing the likelihood of Type I errors, enhancing the reliability of the results.

To explore relationships between continuous variables, Spearman’s correlation coefficient was utilized, chosen for its robustness against non-linear relationships and non-normally distributed data. Statistical significance was rigorously defined at a two-tailed level of 0.05, ensuring a balanced approach to hypothesis testing. All analyses were conducted using SPSS version 26.0, a standard and reliable statistical software in health research.

## 3. Results

### 3.1. Demographics and Professional Characteristics and Details Related to the Specialized Training Received for the Care of Roma Women

The sample consists of 100 participants who represent 30.4% of the employed midwives and obstetricians/gynecologists in Western Greece, with an average age of 45 years (SD = 9.7 years). The sample was 87% female, with only one participant not having Greek nationality. Furthermore, 81% were working as midwives or in related roles, and 63% had completed higher education. Ninety-six per cent of the participants received their training in Greece. All participants held employment, with 55.2% based in Patras, 19.8% in Agrinio, 12.5% in Messolonghi, and 12.5% in Preveza. The majority of them (96%) worked in a hospital’s maternity/gynecology department.

Only 7.1% of Roma women had received specialized training in their care. Eighty-five point seven per cent of the participants had received information about the specific needs of Roma women, their rights, and the stereotypes associated with them. Among those who lacked information, 47.8% expressed a strong desire for information, 18.9% expressed a desire for information under specific conditions, and 10% expressed no desire for information. Furthermore, 83.3% of the participants, who had not received specialized training, expressed a desire to learn about the unique needs of Roma women, while 67.9% expressed a desire to understand their rights.

Appendix A summarizes the sociodemographic characteristics of participants, employment details, and details regarding the receipt of specialized training for the care of Roma women.

### 3.2. Barriers and Challenges Encountered in Receiving Care from the Perspective of the Roma Women

#### 3.2.1. Personal Evaluation of Participants’ Abilities to Provide Adequate Counseling/Care to Roma Women and the Influence of These Abilities

Appendix A presents a personal evaluation of participants’ abilities to provide counseling and care to Roma women. The first question is rated on a 10-point scale (1 = “Not at all” to 10 = “Extremely”), while subsequent questions use a 5-point Likert scale (1 = “Not at all” to 5 = “Extremely” or “Absolutely”). Higher scores indicate a greater perceived ability or stronger agreement with the corresponding statements.

#### 3.2.2. Responses Regarding the Provision of Healthcare Services to Roma, with a Focus on Roma Women

Appendix A presents the responses regarding the provision of healthcare services to Roma, with a focus on Roma women. We scaled the responses from 1 to 5, with 1 signifying strong disagreement and 5 signifying strong agreement. Therefore, higher scores indicate greater agreement with the respective opinion expressed.

#### 3.2.3. Frequency of Communication Problems with Roma During the Provision of Care and the Feelings Experienced by Participants When Providing Services to These Women

Appendix A shows the frequency of communication problems with Roma during the provision of care, as well as the feelings experienced by participants when providing services to these women.

In the last six months, 33% of participants provided counseling/care to 1–5 Roma women with whom there was a communication problem. In general, 14% always faced communication problems when providing services to Roma women, and 53% reported that it happened frequently. The most common feelings when a Roma woman visited the participants’ workplace were understanding, empathy, and interest, with mean scores of 3.6 points (SD = 1 point, SD = 0.9 points, and SD = 1 point, respectively). Respect came next, scoring a mean of 3.2 points (SD = 1.1 points), followed by pity at 2.1 points (SD = 1.1 points), aversion and embarrassment at 2 points (SD = 1.1 points), and indifference at 1.7 points (SD = 0.9 points).

#### 3.2.4. Issues for Which Roma Women Visit or Report to the Participants

The issues for which Roma women visit or report to the participants are presented in Appendix A.

A total of 70.7% of participants reported that Roma women visited them for perinatal care, 62.6% due to pregnancy complications, and 45.5% due to postpartum complications. Additionally, 35.4% mentioned that they received visits due to concerns about unwanted pregnancy, and the same percentage was for sterilization purposes. Visits for information on pregnancy-related issues (30.3%) and accessing financial support (26.3%) were less frequent. Injuries accounted for 24.2% of visits, while problems related to childcare, breastfeeding, domestic violence, and prevention accounted for 15.2%. A total of 14.1% visited to vaccinate their children or were dealing with health problems. Only 11.1% wanted information on contraceptive options, while 10.1% were concerned about hereditary diseases, and the same percentage sought information on housing issues. A total of 8.1% expressed a desire to enhance their health, 6.1% expressed a desire for vaccination, and only 4% expressed concern about managing psychological issues.

#### 3.2.5. Problems That Healthcare Professionals Face During the Provision of Their Services and the Resources Participants Have Access to in Order to Address These Issues

Appendix A provides data on the challenges faced by healthcare professionals in providing their services, as well as the resources available to participants to address these issues.

A total of 72.9% of participants reported that a significant problem was that the Roma women could not understand the information conveyed by healthcare professionals. Additionally, 41.7% mentioned that they were unable to help because the women did not have the necessary documents, and 24% reported that they could not understand what the women were saying. To better serve these patients, 46.6% of participants had access to guidelines from a public agency, 25% had access to tools that facilitated communication, 22.7% had access to guidelines from a scientific organization, and 18.2% had access to resources from another organization (e.g., an NGO).

#### 3.2.6. Evaluation of the Barriers That Roma People Face in Accessing Healthcare Services

Appendix A provides an evaluation of the barriers that Roma people face in accessing healthcare services, presented in descending order. We conducted the evaluation on a scale from 1 to 5, where 1 represents “not at all” and 5 represents “very much”. Higher scores indicate more frequent or severe barriers.

### 3.3. Barriers and Challenges Encountered in Receiving Care from the Perspective of Providers

#### 3.3.1. Main Barriers That Healthcare Professionals Face When Providing Healthcare Services to Roma

Appendix A presents the main barriers that healthcare professionals face when providing healthcare services to Roma, listed in descending order. The most significant problem, chosen by 85% of participants, was that Roma do not comply with the rules, often using emergency departments instead of local healthcare units (TOMY), community clinics, and health centers. A total of 82% identified visits outside of working hours and unscheduled appointments as major issues, while 76% cited illiteracy. For 67%, urban and civic issues were a problem, while 53% pointed to a lack of continuous care, and 35% mentioned language barriers. Furthermore, 29% identified a lack of knowledge about cultural differences, discrimination against Roma, and a lack of mediators, especially in emergency situations, as significant barriers, while 28% considered fear to be a problem.

#### 3.3.2. Information About the Services and Professionals with Whom the Participants Collaborate to Provide Support/Health Services to Roma

Appendix A provides information about the services and professionals with whom the participants collaborate to provide support/health services to Roma. A total of 51.6% collaborated with hospitals, 36.8% with the police, and 30.5% with municipal social services. Health centers reported the lowest rates of collaboration (23.2%), followed by community centers with a Roma branch (21.1%), and TOMY (20%). Following these, 12.6% collaborated with community centers, and 6.3% collaborated with mental health services and private doctors.

Regarding professionals, the most frequent collaboration was with social workers (50.5%), followed by nurses/health visitors (42.3%), and Roma mediators (25.8%). Additionally, there was collaboration with administrative staff (18.6%), psychologists (9.3%), and psychiatrists (6.2%).

#### 3.3.3. Frequency with Which Participants Collaborate with Other Professionals in Case of a Problem

Appendix A presents the frequency with which participants collaborate with other professionals in case of a problem. A total of 36.1% frequently to always used a Roma mediator, 45.1% collaborated with a social worker, 34.9% involved the police, 15% worked with a psychologist, and 11.7% consulted a psychiatrist.

#### 3.3.4. Agreement/Disagreement Rates of Healthcare Professionals Concerning Various Opinions About the Roma

Appendix A shows the agreement/disagreement rates of healthcare professionals concerning various opinions about the Roma. A total of 23.5% believed that Roma women know enough about the organization of healthcare. Significantly higher, 71.7% agreed with the opinion that Roma women have a “theatrical” way of expressing pain, and 68.7% agreed with the view that they do not want to undergo medical tests.

#### 3.3.5. Why Participants Believe Roma Women Do Not Seek Help from Healthcare Services

Appendix A presents the reasons why participants believe Roma women do not seek help from healthcare services, listed in descending order. The most common reason, cited by 57.3% of participants, was that Roma women do not know where to go for their problems. This was followed by 29.2% who believed that they are unsure whether they would receive appropriate care, 20.8% that they fear the reactions of healthcare professionals, 11.5% that they do not trust healthcare professionals, and 10.4% who believed they fear the reactions of other patients. Additionally, 25% thought there were other reasons, with the main one being indifference.

#### 3.3.6. Knowledge

The participating healthcare professionals were asked some knowledge-based questions. Their responses are provided in the following Table 1.

A total of 42% correctly answered that Roma women are not at a higher risk of obesity, 83% knew that detecting cases of violence among pregnant Roma women faces many challenges due to cultural customs, and 91% knew that contraceptive methods should not be discussed only when the woman asks the healthcare professional. Only 21.1% correctly chose the option ‘False’ for the statement that Roma women have higher morbidity rates. Additionally, the correct answer rates were low regarding child mortality in Roma populations, which is not elevated among them. Even lower were the correct answer rates, at 12.1%, concerning the comparison of smoking rates among pregnant Roma women versus non-Roma women and the overall pregnant population. On the other hand, the majority, 82.8%, correctly answered that Roma are Greek citizens, 73.7% that they have the same opportunities for access to health services, 86.7% that it is the responsibility of Roma mediators to communicate with the midwife for the care of the pregnant and postpartum Roma woman, 91.8% that the mediators are also responsible for communicating with the social worker on social protection issues, and 89.8% that they are also responsible for communication with the teacher at the school of the pregnant Roma teenager.

Subsequently, a knowledge score was calculated from the above responses, assigning a value of 1 for each correct answer and 0 for each incorrect one. This was converted into a percentage score. Higher values indicate more knowledge.

In this particular sample, the minimum knowledge score percentage, recorded by 4% of the sample, was 25 points, and the maximum, recorded by 1%, was 100 points, with an average score of 59.2 points (SD = 15.5 points).

### 3.4. Frequencies of Various Problems Occurring Among Roma Women Compared to Non-Roma Women According to the Participant’s Answers

Table 2 presents the frequencies of various problems occurring among Roma women compared to non-Roma women.

A total of 48.9% believed that preterm births are more frequent among Roma women compared to others, 32% believed that they more often experience complications, and 39.3% believed that they consume alcohol during pregnancy. Additionally, according to 67%, it was more likely for a Roma woman to be pregnant without a partner. The majority (97.9%) considered it more common for these women to have insufficient gynecological monitoring, 86.7% thought they have financial problems, 66.7% believed they face domestic violence, and 96% believed they experience pregnancies during adolescence. Only 22.6% considered depression more common among them, while 97.0% believed they more frequently have a low educational level.

### 3.5. Knowledge Score of the Participants Based on Their Demographic and Employment Characteristics

Table 3 shows the knowledge score of the participants based on their demographic and employment characteristics.

The knowledge score was found to differ based on profession and educational level. Specifically, obstetricians/gynecologists had higher scores, and therefore more knowledge, compared to midwives (*p* = 0.037). Additionally, those with a master’s or doctoral degree had more knowledge compared to those with only a tertiary education degree (*p* = 0.022). Finally, after Bonferroni corrections, no significant difference in the knowledge score was found based on the place of employment.

Furthermore, no significant correlation was found between age and the knowledge score, as the Spearman correlation coefficient was rho = 0.16 (*p* = 0.115).

### 3.6. Correlation of Cultural Competence Score with Demographic Characteristics

A total score was calculated from the first three questions regarding cultural competence, as the mean of these questions. The score ranges from 1 to 5 points, with higher values indicating greater cultural competence and the ability to provide adequate counseling/care to Roma women. In this particular sample, the minimum score was 1.5 points and the maximum was 5 points, with a mean score of 3.5 points (SD = 0.7 points).

Table 4 presents the cultural competence score of the participants based on their demographic and employment characteristics.

The cultural competence score was not found to differ based on the aforementioned demographic and employment characteristics of the participants.

The relationship between the cultural competence score and age was then examined. Older age was associated with a higher cultural competence score (rho = 0.20, *p* = 0.048), indicating a greater ability to provide adequate counseling/care to Roma women.

Figure 1 shows the correlation between age and the cultural competence score.

### 3.7. Relationship Between the Cultural Competence Score and Knowledge Score

The relationship between the cultural competence score and knowledge score was then examined. Greater knowledge was associated with a higher cultural competence score (rho = 0.34, *p* = 0.001), indicating a greater ability to provide adequate counseling/care to Roma women.

Figure 2 shows the correlation between knowledge and the cultural competence score.

### 3.8. Relationship Between the Provision of Healthcare Services to Roma and the Professional Position of the Participants

The relationship between the provision of healthcare services to Roma and the professional position of the participants was then examined (Table 5).

Obstetricians/gynecologists had a higher score than midwives (*p* = 0.033), indicating that they believed to a greater extent that Roma have the same rights to access healthcare services as other citizens. They also believed to a greater extent that more of their colleagues do not know how to provide care/support to Roma (*p* < 0.001).

Figure 3 illustrates the agreement score for the question regarding the rights of access to healthcare services by professional category.

Figure 4 illustrates the score for the question regarding knowledge of providing care/support to Roma by professional category.

### 3.9. Relationship Between Profession and the Feelings Towards Roma Women and Evaluation of the Barriers That Participants Believe Roma Face in Accessing Healthcare Services

The relationship between profession and the feelings towards Roma women, as well as the evaluation of the barriers that participants believe Roma face in accessing healthcare services, was examined. For this purpose, a feelings score was calculated as the average of all feelings (negative feelings were reversed), a total score was calculated for the overall evaluation of all barriers faced by Roma, and a total score was calculated as the sum of the barriers faced by healthcare professionals themselves. Higher values in each score correspond to more positive feelings towards Roma patients, more frequent occurrence/greater severity of a barrier, and more difficulties faced by healthcare professionals in dealing with Roma patients, respectively (Table 6).

The participants’ feelings, the evaluation of the barriers faced by Roma, and the problems faced by healthcare professionals themselves did not differ based on their professional position.

## 4. Discussion

The findings of this study highlight several important aspects related to the cultural competence of maternity/healthcare professionals providing care to Roma women in Western Greece. The moderate to high levels of cultural competence observed among the study participants are consistent with the existing literature, which emphasizes the importance of cultural competence in enhancing healthcare outcomes for marginalized populations. This is particularly crucial in the context of Roma women, who often face multiple barriers to accessing healthcare, including discrimination, communication challenges, and a lack of understanding of their unique cultural needs [12]. The healthcare challenges experienced by Roma women stem from both systemic barriers and interpersonal issues. Low health literacy and socioeconomic disadvantages limit their access to preventive care and appropriate services, while cultural gaps often hinder effective communication between patients and providers. Healthcare professionals frequently report difficulties such as incomplete documentation and unscheduled visits, which are compounded by insufficient training in cultural competence [13].

In the present study, midwives and obstetricians/gynecologists reported that their predominant feelings towards Roma women were understanding, empathy, and concern. This contrasts with the findings of the Roma themselves in the study by Goupou [14], where they reported that healthcare professionals exhibit negative attitudes and harbor prejudices against them. Regarding the professional position of the participants, it was found that there were no differences in the evaluation of the barriers faced by Roma, the feelings of the participants, and the problems faced by healthcare professionals themselves when providing care to Roma women [14].

Colombini et al. [15] also investigated the barriers to access for Roma in Southeast Europe to sexual and reproductive health services through focus group discussions with a total of 58 participants from Roma communities in Albania, Bulgaria, and FYROM. Their findings revealed numerous barriers for Roma in accessing sexual and reproductive health services, the most significant being the overall lack of financial resources, healthcare providers’ demands for informal payments, lack of health insurance, and geographical obstacles. Thus, it was found that the healthcare systems in these countries do not provide financial protection and equitable services to Roma women, and therefore, it is suggested that awareness and information be improved to address these inequalities. Rusi [16] also points out that Roma women in Albania are excluded in various ways from accessing reproductive health services due to economic, geographical, informational, and institutional barriers. Roma women also face other issues, such as lack of access to public maternity services due to lack of documentation, lack of information, and low awareness of Roma women about the components of reproductive health and family planning.

Kovalčíková and Kresila’s study explored cultural competence through an examination of adaptive skills among Roma children, providing insights into the distinct cultural expectations within Roma families. Their findings revealed a significant emphasis on gender-specific competencies, with girls being trained early in caregiving roles such as sibling care, personal hygiene, and assisting with household tasks, while boys are guided toward independent subsistence activities. These culturally defined roles underscore the importance of understanding cultural contexts in designing educational and social interventions. Such insights highlight the need for culturally sensitive approaches that respect community-specific values while promoting equitable access to education and healthcare [17]. In the present study, the investigation of the barriers encountered by Roma in accessing healthcare services highlighted as the most significant problem the lack of health literacy and prevention, the difficulty in understanding, and diagnostic tests. At the same time, the participants identified the most significant barriers faced by healthcare professionals when providing health services to Roma as the fact that Roma do not comply with rules and use emergency departments instead of TOMY, health centers, and Primary Care Units, while a large percentage make visits outside working hours, and appointments are unscheduled. Illiteracy also emerged as a significant problem. Thus, according to the participants, in order to provide health services to Roma, they collaborate with the hospital authorities to which they belong and secondarily with the police and municipal services. In cases where assistance was needed, they turned to a Roma mediator. Particularly in the case of Roma women facing problems, the participants highlighted the significant issue of awareness, as most reported that Roma women do not know where to turn for their problems.

Consequently, the issue of communication and understanding of instructions is highlighted in other studies, as well as in the present one. However, in this particular study, the participants noted that Roma women are partly facilitated in their communication with healthcare professionals providing maternity care as there is access to relevant public bodies, which they approach either on their own or with the mediation of the above-mentioned healthcare professionals.

The level of cultural competence of the participants was found to be high, especially in areas concerning the provision of counseling and care to Roma women in health promotion, referrals for social care, and addressing cultural differences. This contrasts with the literature review by Watson and Downe [18], which investigated ten studies on the experiences of Roma women with discrimination in maternity care in Europe and any interventions to address them. This study found that many Roma women face barriers to accessing maternity care. Even when they can access care, they may experience mistreatment, such as discrimination based on ethnicity, economic status, place of residence, or language. Therefore, in the studies included in this review, the cultural competence of midwives is considered not as high as in the present study, where the participants possess cultural competence at a high level, have been appropriately informed about providing maternity and counseling care to Roma women, and emphasize that they feel understanding, empathy, and concern for Roma women.

A significant finding from this study is the correlation between the level of education and cultural competence. Healthcare professionals providing maternity care with higher educational qualifications, particularly those with postgraduate degrees, demonstrated greater cultural competence. Moreover, the study identified that older healthcare professionals exhibited higher levels of cultural competence. This could be attributed to their extensive experience and the likelihood of having encountered diverse patient populations over the years [19].

Addressing the barriers identified in accessing healthcare services, particularly for marginalized communities, requires not only understanding but also actionable strategies to improve cultural competence among healthcare professionals. For instance, integrating cultural competence training into healthcare curricula and continuing education programs can enhance the ability of providers to engage effectively with diverse populations. This training should focus on practical components such as intercultural communication, managing biases, and understanding specific community needs. For example, Roma healthcare mediators have been shown to bridge communication gaps and foster trust between healthcare providers and Roma patients, suggesting the need for similar community-specific interventions.

Further, targeted measures such as employing culturally adapted materials, providing multilingual resources, and enhancing the accessibility of services can help mitigate structural barriers. Healthcare organizations could benefit from adopting systematic policies to address discrimination and promote inclusivity. Practical suggestions, including improving health literacy through community-based workshops and leveraging technology for better service delivery, can also reduce disparities. These efforts, when combined, hold the potential to improve healthcare access and outcomes while fostering equitable and culturally sensitive care.

### 4.1. Limitations

This study has several limitations that should be acknowledged. First, the use of convenience sampling may have introduced selection bias, limiting the generalizability of the findings to all healthcare providers in Western Greece. Another limitation of this study is the relatively small sample size, which, while representing 30.4% of the employed midwives and obstetricians/gynecologists in Western Greece, may not fully capture the diversity of experiences and perspectives of all healthcare professionals in the region. Additionally, the use of convenience sampling could introduce selection bias, potentially limiting the generalizability of the findings. The lack of an a priori power analysis to determine the sample size needed for sufficient statistical power is another limitation that should be considered. Future studies with larger, more representative samples and robust sampling strategies could provide a more comprehensive understanding of the cultural competence of healthcare providers in this context. Additionally, the cross-sectional design of the study does not allow for the examination of changes in cultural competence over time. The reliance on self-reported data from questionnaires may also have resulted in social desirability bias, where participants may have over-reported their cultural competence. Lastly, the study focused on a specific geographical region, which may limit the applicability of the findings to other contexts or regions with different demographic characteristics.

Despite these limitations, the findings provide valuable insights into the cultural competence of healthcare providers, particularly in addressing the needs of Roma women. While extrapolation should be approached cautiously, the study identifies trends and challenges that are likely applicable in similar multicultural or underserved populations, offering a foundation for future research and targeted interventions.

### 4.2. Future Considerations

Future research should consider longitudinal studies to assess the development of cultural competence over time among maternity/healthcare providers. Crucially, studies should aim to include larger and more diverse sample groups to enhance the generalizability of findings and better represent the broader population. Expanding the scope of research to include a broader range of regions and incorporating qualitative methods, such as interviews or focus groups with Roma women, could provide a more comprehensive understanding of the barriers they face and the effectiveness of cultural competence training. Additionally, exploring the impact of targeted educational programs on improving healthcare outcomes for Roma women could inform the development of more effective interventions. Efforts should also be made to engage Roma communities in the design and implementation of healthcare initiatives to ensure that their needs and perspectives are adequately addressed.

Future initiatives should prioritize the integration of cultural competence education into the curricula of medical, nursing, and midwifery schools. This training should include practical modules on intercultural communication, addressing biases, and understanding the unique needs of marginalized groups such as the Roma community. Incorporating experiential learning opportunities, such as community engagement projects and simulations, can enhance the ability of future healthcare providers to deliver culturally sensitive care. Such measures would ensure that emerging healthcare professionals are better equipped to address health disparities and foster trust with diverse populations.

## 5. Conclusions

The findings of this study highlight the critical role of cultural competence—encompassing knowledge, skills, attitudes, and awareness—in improving healthcare delivery for Roma women in Western Greece. Healthcare providers with greater knowledge of Roma cultural, social, and healthcare challenges exhibited higher levels of cultural competence, emphasizing the need for practical, targeted interventions. These include integrating structured cultural competence training into medical, nursing, and midwifery curricula, implementing continuous professional development programs such as workshops and community immersion activities, and expanding the role of Roma healthcare mediators to bridge communication gaps and build trust. Policy reforms at the institutional level should promote inclusive care and address systemic barriers, while community engagement efforts can foster collaboration to design healthcare initiatives tailored to the needs of Roma women. By addressing key issues such as health literacy, communication challenges, and systemic inequities, these practical measures can reduce healthcare disparities, improve patient–provider relationships, and enhance health outcomes for Roma women. Future research should focus on evaluating the long-term impact of such interventions to ensure sustainable improvements in culturally sensitive healthcare delivery.

## Figures and Tables

**Figure 1 healthcare-13-00190-f001:**
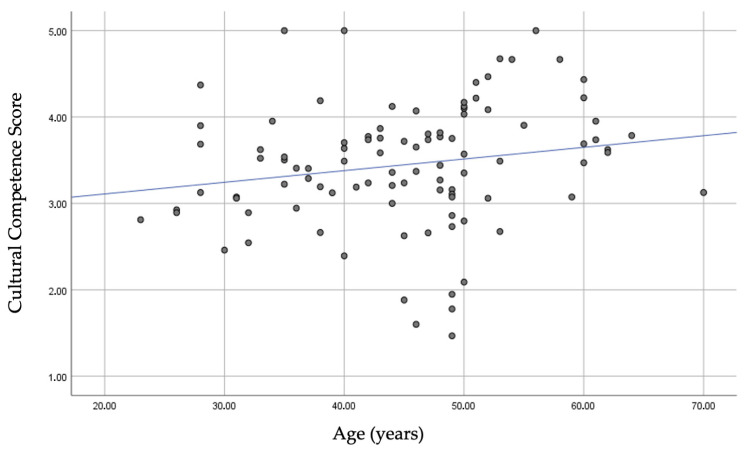
Correlation between cultural competence score and the age of the participants.

**Figure 2 healthcare-13-00190-f002:**
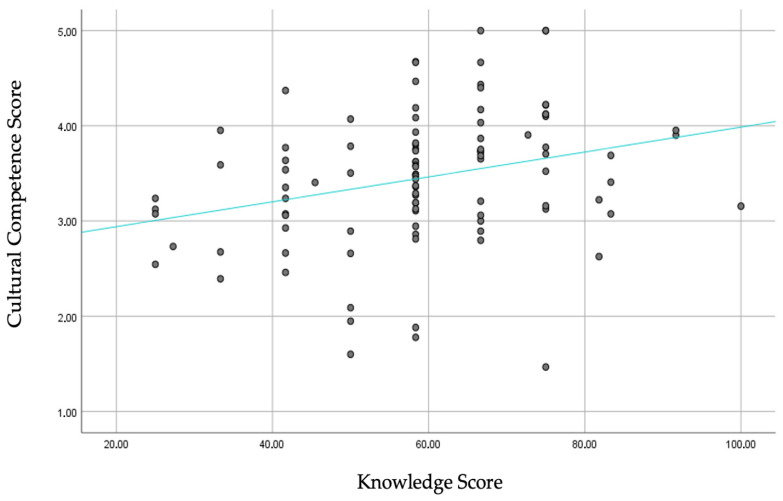
Correlation between knowledge and the cultural competence score.

**Figure 3 healthcare-13-00190-f003:**
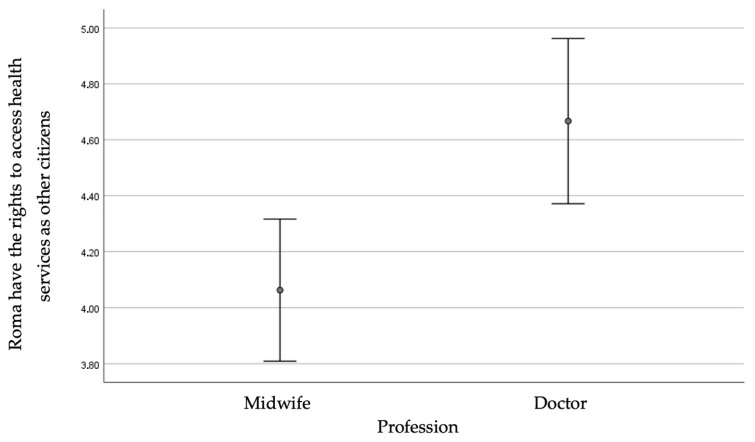
Agreement scores for the question regarding the rights of access to healthcare services by professional category.

**Figure 4 healthcare-13-00190-f004:**
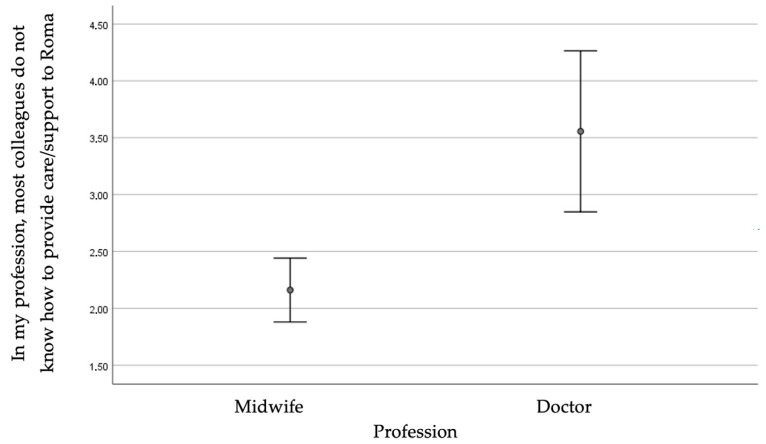
Score for the question regarding knowledge of providing care/support to Roma by professional category.

**Table 1 healthcare-13-00190-t001:** Participants’ responses to knowledge questions.

Question	Ν	%	Correct Answers
Roma women are at higher risk of obesity			
True	19	19.0	42.0
False	42	42.0	
Don’t know	39	39.0	
Detection of violence cases in pregnant Roma women faces many difficulties due to cultural customs			
True	83	83.0	83.0
False	8	8.0	
Don’t know	9	9.0	
Roma women exhibit higher morbidity rates.			
True	46	48.4	21.1
False	20	21.1	
Don’t know	29	30.5	
Contraceptive methods should only be discussed when the woman asks the healthcare professional			
True	8	8.0	91.0
False	91	91.0	
Don’t know	1	1.0	
Infant mortality rates are higher in Roma populations			
True	32	32.3	20.2
False	20	20.2	
Don’t know	47	47.5	
Smoking rates are higher among pregnant Roma women compared to the general population of non-Roma pregnant women			
True	65	65.7	12.1
False	12	12.1	
Don’t know	22	22.2	
Smoking rates are higher among pregnant Roma women compared to the general pregnant population			
True	63	63.6	12.1
False	12	12.1	
Don’t know	24	24.2	
Roma are Greek citizens			
True	82	82.8	82.8
False	4	4.0	
Don’t know	13	13.1	
Roma have the same opportunities to access healthcare services			
True	73	73.7	73.7
False	17	17.2	
Don’t know	9	9.1	
It is the responsibility of Roma mediators to communicate with the midwife about the care of pregnant and postpartum Roma women			
True	46	46.9	86.7
False	13	13.3	
Don’t know	39	39.8	
It is the responsibility of Roma mediators to communicate with the social worker about social protection issues			
True	55	56.1	91.8
False	8	8.2	
Don’t know	35	35.7	
It is the responsibility of Roma mediators to communicate with the teacher at the school of the pregnant Roma teenager			
True	47	48.0	89.9
False	10	10.2	
Don’t know	41	41.8	

**Table 2 healthcare-13-00190-t002:** Frequency of problems in Roma women compared to non-Roma women.

Issue	Very Often N (%)	Often N (%)	As Often as Non-Roma N (%)	Rarely N (%)	Key Implications
Preterm labor	10 (10.4%)	37 (38.5%)	43 (44.8%)	6 (6.3%)	High prevalence suggests potential gaps in prenatal care
Complicated labor	3 (3.1%)	28 (28.9%)	56 (57.7%)	10 (10.3%)	Requires tailored interventions during delivery
Alcohol consumption during pregnancy	5 (5.3%)	32 (34.0%)	27 (28.7%)	30 (31.9%)	Indicates the need for substance use education
Pregnancy without a partner	32 (32.0%)	35 (35.0%)	18 (18.0%)	15 (15.0%)	Highlights socio-economic vulnerabilities
Inadequate gynecological monitoring	64 (64.6%)	33 (33.3%)	2 (2.0%)	0 (0.0%)	Severe gap necessitating targeted interventions
Financial problems	46 (46.9%)	39 (39.8%)	10 (10.2%)	3 (3.1%)	Financial barriers remain a critical challenge
Domestic violence	22 (22.9%)	42 (43.8%)	28 (29.2%)	4 (4.2%)	Reflects urgent need for psychosocial support services
Teenage pregnancy	75 (75.0%)	21 (21.0%)	0 (0.0%)	4 (4.0%)	Indicates an educational and healthcare gap.
Depression	8 (8.2%)	14 (14.4%)	48 (49.5%)	27 (27.8%)	Undetected or unreported cases may be an issue
Low educational level	78 (78.0%)	19 (19.0%)	1 (1.0%)	2 (2.0%)	Significant educational disadvantage impacting health literacy

**Table 3 healthcare-13-00190-t003:** Correlation of knowledge score with demographic characteristics.

Demographic Characteristic		Knowledge Score		*p*
		Mean (SD)	Median (IQR)	
Gender	Female	58.3 (15)	58.3 (50–66.7)	0.132 +
	Male	65.3 (17.5)	66.7 (58.3–81.8)	
Profession	Midwife	57.6 (15.3)	58.3 (50–66.7)	0.037 +
	Doctor	65.9 (14.8)	66.7 (58.3–81.8)	
Education	Up to Tertiary education	56.7 (16.4)	58.3 (41.7–66.7)	0.022 +
	MSc/PhD	64.6 (11.5)	66.7 (58.3–75)	
Workplace	Patras	54.9 (15.8)	58.3 (41.7–66.7)	0.031 ++
	Agrinio	63.3 (14.2)	66.7 (58.3–75)	
	Messolonghi	64.5 (9.9)	58.3 (58.3–75)	
	Preveza	63.9 (11.4)	62.5 (58.3–66.7)	

+ Mann–Whitney test; ++ Kruskal–Wallis; MSc: master’s degree; PhD: Doctorate; SD: standard deviation; IQR: interquartile range.

**Table 4 healthcare-13-00190-t004:** Correlation of cultural competence score with demographic characteristics.

Demographic Characteristic		Cultural Competence Score		*p*
		Mean (SD)	Median (IQR)	
Gender	Female	3.4 (0.7)	3.5 (3.1–3.9)	0.787 +
	Male	3.5 (0.5)	3.6 (3.1–3.8)	
Profession	Midwife	3.4 (0.8)	3.5 (3.1–3.9)	0.655 +
	Doctor	3.5 (0.5)	3.6 (3.1–3.8)	
Education	Up to Tertiary education	3.4 (0.8)	3.4 (3.1–3.9)	0.609 +
	MSc /PhD	3.5 (0.5)	3.7 (3.1–3.8)	
Workplace	Patras	3.3 (0.7)	3.3 (2.9–3.7)	0.081 ++
	Agrinio	3.7 (0.7)	3.7 (3.2–4.2)	
	Messolonghi	3.4 (0.9)	3.7 (2.9–4.1)	
	Preveza	3.7 (0.5)	3.6 (3.4–3.9)	

+ Student’s *t* test; ++ ANOVA; MSc: master’s degree; PhD: Doctorate; SD: standard deviation; IQR: interquartile range.

**Table 5 healthcare-13-00190-t005:** Association of health service provision to Roma with the profession of participants.

Parameter	Profession				*p*
	Midwife		Doctor		
	Mean (SD)	Median (IQR)	Mean (SD)	Median (IQR)	
Roma have the same rights to access health services as other citizens	4.1 (1.1)	4 (3–5)	4.7 (0.6)	5 (4–5)	0.033 +
Roma should be hospitalized in separate wards	3.2 (1.4)	3 (2–5)	3.1 (1.7)	3 (1–5)	0.783 +
Prejudices against Roma are a serious social problem	3.7 (1.2)	4 (3–5)	3.9 (1.3)	4.5 (3–5)	0.316 +
Roma women should be examined by Roma health professionals	1.8 (1.0)	1 (1–2)	1.6 (1.0)	1 (1–2)	0.479 +
Roma women should receive support services from Roma professionals	2 (1.2)	2 (1–3)	1.6 (1.0)	1 (1–2)	0.172 +
As a professional, I would definitely provide care/support to a Roma woman	4.4 (0.8)	5 (4–5)	4.4 (0.8)	5 (4–5)	0.950 +
In my profession, I can easily tell if someone is Roma	4.5 (0.7)	5 (4–5)	4.6 (0.8)	5 (4–5)	0.608 +
In my profession, most colleagues do not know how to provide care/support to Roma	2.2 (1.3)	2 (1–3)	3.6 (1.4)	4 (3–5)	<0.001 +
I recognize the social needs of the Roma	3.6 (1.1)	4 (3–4)	3.9 (0.8)	4 (3–5)	0.245 +
Most health issues of Roma women are due to their cultural peculiarities	3.8 (1.1)	4 (3–5)	3.9 (0.9)	4 (3–5)	0.580 +

+ Mann–Whitney test; SD: standard deviation; IQR: interquartile range.

**Table 6 healthcare-13-00190-t006:** Association of profession with emotional handling of Roma women and presentation of barriers regarding the provision of health services.

Parameter	Profession				*p*-Value
	Midwife		Doctor		
	Mean (SD)	Median (IQR)	Mean (SD)	Median (IQR)	
Score of Positive Emotions	3.8 (0.7)	3.3 (3–3.8)	3.5 (0.8)	3.7 (3.3–4.1)	0.341 +
Score of Barrier Assessment	3.3 (0.6)	3.3 (3–3.8)	3.5 (0.8)	3.6 (3.3–4.1)	0.084 +
Score of Difficulties with Roma Patients	5.2 (2.4)	5.0 (4–7)	4.7 (2.6)	4.0 (3–6)	0.201 +

+ Mann–Whitney test; SD: standard deviation; IQR: interquartile range.

## Data Availability

The data presented in this study are available within the article.

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
