# Peer review of "Cultural Competence of Obstetricians/Gynecologists and Midwives Providing Midwifery Care to Roma Women in Western Greece"

_healthcare, 2025, doi:10.3390/healthcare13020190_

Round 1
Reviewer 1 Report
Comments and Suggestions for Authors
We thank the authors for highlighting such an important topic. Here are my comments:
1. Please elaborate on the challenges faced by the Roma community. Also define Roma community. Who are they? What is the origin of this population? How much do they make up of the Greek population? Also, mention health literacy as a major barrier here as they go hand in hand with cultural competence.
2. In line 70, replace "physical approach" with "in-person".
3. In line 78, can you elaborate on the previous study using this questionnaire? Is the questionnaire validated?
4. The results should be condensed intro three sub-sections: 1. demographics; 2. barriers and challenges encountered in receiving care from the perspective of the roma women; and 3. barriers and challenges encountered in receiving care from the perspective of providers.
5. In the future considerations section, what about training the next generation of health professionals to be more culturally competent?
6. For figures 3 and 4, how was the agreement score calculated? the y-axis is not clear.
Author Response
Reviewer 1
Comments and Suggestions for Authors
We thank the authors for highlighting such an important topic. Here are my comments:
- Please elaborate on the challenges faced by the Roma community. Also define Roma community. Who are they? What is the origin of this population? How much do they make up of the Greek population? Also, mention health literacy as a major barrier here as they go hand in hand with cultural competence.
AUTHORS’ RESPONSE: WE added this data: The Roma community, often referred to as Romani, is an ethnic minority with origins tracing back to the Indian subcontinent. Linguistic and genetic evidence suggests that they migrated to Europe over a millennium ago, during the early medieval period. Today, they are one of the largest and most dispersed ethnic minorities in Europe, with significant populations across various countries. The Roma people have a rich cultural heritage characterized by unique traditions, languages (such as Romani), and a nomadic history [4, 5].
In Greece, the Roma are an integral but marginalized part of the population. Estimates suggest that they comprise approximately 2-3% of the Greek population. While exact numbers vary due to a lack of consistent census data, it is estimated that there are about 200,000–300,000 Roma individuals in Greece.
The Roma community faces significant challenges, including profound health disparities driven by socioeconomic disadvantages, such as poverty, inadequate housing, and limited access to clean water, which contribute to higher rates of disease and poor maternal and child health outcomes. Health literacy is a critical barrier, with limited knowledge about healthcare rights and preventive services leading to underutilization and difficulty navigating healthcare systems. Cultural competence gaps among healthcare providers often result in misinterpretation of Roma cultural practices, reinforcing stereotypes and mistrust. Discrimination and stereotyping, both at an individual and institutional level, further marginalize Roma individuals, while structural barriers, such as the lack of Roma mediators and bureaucratic challenges, continue to restrict their access to equitable healthcare services [7].
- In line 70, replace "physical approach" with "in-person".
AUTHORS’ RESPONSE: We made the correction. Thank you.
- In line 78, can you elaborate on the previous study using this questionnaire? Is the questionnaire validated?
AUTHORS’ RESPONSE: We added this: The researchers in that study investigated the healthcare access and challenges faced by Roma women in Greece, focusing on sexual and reproductive health, using mixed methods that included questionnaire that was validated for healthcare and other professionals, as well as interviews and focus groups, which revealed themes like stereotypes, cultural communication barriers, and the need for systemic changes and Roma mediators
- The results should be condensed intro three sub-sections: 1. demographics; 2. barriers and challenges encountered in receiving care from the perspective of the roma women; and 3. barriers and challenges encountered in receiving care from the perspective of providers.
AUTHORS’ RESPONSE: We structured the Results section as you suggested.
- In the future considerations section, what about training the next generation of health professionals to be more culturally competent?
AUTHORS’ RESPONSE: Future initiatives should prioritize the integration of cultural competence education into the curricula of medical, nursing, and midwifery schools. This training should include practical modules on intercultural communication, addressing biases, and understanding the unique needs of marginalized groups such as the Roma community. Incorporating experiential learning opportunities, such as community engagement projects and simulations, can enhance the ability of future healthcare providers to deliver culturally sensitive care. Such measures would ensure that emerging healthcare professionals are better equipped to address health disparities and foster trust with diverse populations.
- For figures 3 and 4, how was the agreement score calculated? the y-axis is not clear.
AUTHORS’ RESPONSE: The agreement scores in Figures 3 and 4 represent participants' levels of agreement with statements regarding the provision of healthcare to Roma women. These scores were calculated using a Likert scale format, where participants rated their agreement with various statements. The y-axis represents the mean score of agreement across participants, where higher values indicate greater agreement with the statement.
For Figure 3, the agreement score reflects responses to the statement about Roma individuals having equal rights to healthcare access. For Figure 4, the score corresponds to participants' self-assessed knowledge and ability to provide care/support to Roma women. The specific scale (e.g., 1–5 or 1–10) is not explicitly clarified in the legend but can be inferred from similar sections in the methodology and results where Likert scales were used to quantify agreement.
Reviewer 2 Report
Comments and Suggestions for Authors
Thank you for inviting me to review. The research topic is important. However, I have a few suggestions that could improve the quality of the manuscript.
Introduction
Explaining cultural competences based on one item is insufficient and requires supplementation, including providing various sources.
Sampling
Please consider adding what the convenience sampling method was.
Data collection
Please consider adding how data collection was carried out.
Research tool
REACH project questionnaire questions - what does this acronym mean
Lines 86-88, 144-147 unclear
„The questions in the first and second sections are mostly in the form of a 5-point Likert scale ranging from "Not at all" to "Extremely" others from "Not at all" to "Absolutely," and others from "Strongly disagree" to "Strongly agree." The questions for collecting demographic data are multiple-choice.
Supplementary Table 2 presents a personal evaluation of participants' abilities to provide adequate counseling/care to Roma women, as well as the influence of these abilities. Participants rate the first question on a scale from 1 to 10, and the subsequent questions on a scale from 1 to 5. A higher score indicates a greater ability or stronger agreement with the respective statement.
Study group
Small, no inclusion or exclusion criteria described, what percentage of employed midwives and obstetricians/gynecologists in Western Greece.
Results
The results section is too long.
Discussion and References
Are there any other studies that could support the discussion and supplement the literature?
Conclusions unclear - what is this knowledge supposed to be about? cultural competences are knowledge, skills, attitudes, awareness
The positive correlation between higher knowledge levels and cultural competence suggests that enhancing educational efforts among healthcare providers can improve care outcomes.
Author Response
Author's Reply to the Review Report (Reviewer 2)
Thank you for inviting me to review. The research topic is important. However, I have a few suggestions that could improve the quality of the manuscript.
Introduction
Explaining cultural competences based on one item is insufficient and requires supplementation, including providing various sources.
AUTHORS RESPONSE: We revised as follows: Cultural competence is increasingly recognized as a cornerstone of effective healthcare delivery, particularly in multicultural societies where patients present diverse ethnicities, languages, traditions, and health beliefs [1]. Defined as "the ability of individuals to establish effective interpersonal and working relationships that supersede cultural differences" [2], cultural competence enables healthcare providers to respect and integrate the cultural, social, and linguistic needs of patients. By addressing the multifaceted social determinants of health and fostering meaningful patient-provider relationships, cultural competence contributes to the reduction of health disparities and the improvement of patient satisfaction and equity in healthcare outcomes [2].
The components of cultural competence include cultural awareness, cultural knowledge, cultural skills, cultural encounters, and cultural desire [3]. These elements collectively promote the ability of healthcare providers to deliver personalized, patient-centered care in diverse settings. Evidence suggests that interventions aimed at enhancing cultural competence improve healthcare providers' knowledge, attitudes, and skills, and positively influence patient satisfaction [3]. However, gaps remain in linking these outcomes to tangible improvements in patient health and adherence, highlighting the need for further research to optimize educational strategies and measure patient-centered outcomes comprehensively
Sampling
Please consider adding what the convenience sampling method was.
AUTHORS’ RESPONSE: This non-probability sampling approach selected individuals who were readily accessible and willing to participate, making it a practical choice for exploratory research. While convenience sampling facilitates data collection in specific contexts, it may limit the generalizability of findings due to potential selection bias.
Data collection
Please consider adding how data collection was carried out.
AUTHORS’ RESPONSE: 2.2. Data Collection Method
The data collection was carried out through an in-person approach at healthcare facilities, including hospitals and health centers, where midwives and obstetricians/gynecologists were employed. Researchers scheduled visits to these locations and approached eligible participants during breaks or at times pre-arranged with department heads to minimize disruption to clinical duties.
Participants were provided with a brief verbal explanation of the study's purpose and relevance, emphasizing the importance of their input in understanding cultural competence in maternity care for Roma women. To ensure participants felt comfortable, researchers explained that responses would remain anonymous, and no identifiable information would be linked to the questionnaire data.
To facilitate convenience and ensure confidentiality, participants were given the option to complete the questionnaire on-site or take it to complete later in a private setting. A secure collection method was established, where completed questionnaires could be placed in sealed envelopes or collected in a locked drop box provided at each facility. The envelopes or drop boxes were retrieved by the research team at designated intervals.
Each participant received an approximate timeframe for questionnaire completion, ensuring they did not feel rushed. Researchers remained available at the collection sites to address any logistical concerns or clarify queries without influencing the participants' responses.
Research tool
REACH project questionnaire questions - what does this acronym mean
AUTHORS’ RESPONSE: We added this: Roma Women’s Empowerment and Fight against discrimination in Access to Health
Lines 86-88, 144-147 unclear
„The questions in the first and second sections are mostly in the form of a 5-point Likert scale ranging from "Not at all" to "Extremely" others from "Not at all" to "Absolutely," and others from "Strongly disagree" to "Strongly agree." The questions for collecting demographic data are multiple-choice.
AUTHORS’ RESPONSE: We revised as follows: The questions in the first and second sections use a variety of Likert scales to assess participants' responses. These include a 10-point scale ranging from "Not at all" to "Extremely" for assessing capability in providing care, and a 5-point Likert scale ranging from "Not at all" to "Absolutely", as well as "Strongly disagree" to "Strongly agree", depending on the question type. Additionally, participants were asked about their perceptions and experiences using multiple-choice questions and categorical responses, such as frequency (e.g., "Always", "Frequently", "Rarely"), and demographic questions to collect information like age, gender, education, and workplace.
Supplementary Table 2 presents a personal evaluation of participants' abilities to provide adequate counseling/care to Roma women, as well as the influence of these abilities. Participants rate the first question on a scale from 1 to 10, and the subsequent questions on a scale from 1 to 5. A higher score indicates a greater ability or stronger agreement with the respective statement.
AUTHORS’ RESPONSE: We revised as follows: Supplementary Table 2 presents a personal evaluation of participants' abilities to provide counseling and care to Roma women. The first question is rated on a 10-point scale (1 = "Not at all" to 10 = "Extremely"), while subsequent questions use a 5-point Likert scale (1 = "Not at all" to 5 = "Extremely" or "Absolutely"). Higher scores indicate a greater perceived ability or stronger agreement with the corresponding statements.
Study group
Small, no inclusion or exclusion criteria described, what percentage of employed midwives and obstetricians/gynecologists in Western Greece.
AUTHORS’ RESPONSE: We added this:
Inclusion criteria were professionals actively involved in providing midwifery or obstetric care to Roma women within the region. Exclusion criteria included healthcare professionals without direct contact or experience working with Roma populations. This ensured that participants had relevant exposure to the context under investigation.
We added also this: The sample consists of 100 participants who represent 30.4% of the employed midwives and obstetricians/gynecologists in Western Greece,
Results
The results section is too long.
AUTHORS RESPONSE: We moved a part of the results to the Supplementary File 2.
Discussion and References
Are there any other studies that could support the discussion and supplement the literature?
AUTHORS’ RESPONSE: We added this:
The healthcare challenges experienced by Roma women stem from both systemic barriers and interpersonal issues. Low health literacy and socioeconomic disadvantages limit their access to preventive care and appropriate services, while cultural gaps often hinder effective communication between patients and providers. Healthcare professionals frequently report difficulties such as incomplete documentation and unscheduled visits, which are compounded by insufficient training in cultural competence [12].
Kovalčíková and Kresila's study explored cultural competence through an examination of adaptive skills among Roma children, providing insights into the distinct cultural expectations within Roma families. Their findings revealed a significant emphasis on gender-specific competencies, with girls being trained early in caregiving roles such as sibling care, personal hygiene, and assisting with household tasks, while boys are guided toward independent subsistence activities. These culturally defined roles underscore the importance of understanding cultural contexts in designing educational and social interventions. Such insights highlight the need for culturally sensitive approaches that respect community-specific values while promoting equitable access to education and healthcare [16]
Conclusions unclear - what is this knowledge supposed to be about? cultural competences are knowledge, skills, attitudes, awareness
The positive correlation between higher knowledge levels and cultural competence suggests that enhancing educational efforts among healthcare providers can improve care outcomes.
AUTHORS’ RESPONSE: We revised the Conclusions section as follows: The findings of this study highlight the critical role of cultural competence—encompassing knowledge, skills, attitudes, and awareness—in improving healthcare delivery for Roma women in Western Greece. Healthcare providers with greater knowledge of Roma cultural, social, and healthcare challenges exhibited higher levels of cultural competence, emphasizing the need for practical, targeted interventions. These include integrating structured cultural competence training into medical, nursing, and midwifery curricula, implementing continuous professional development programs such as workshops and community immersion activities, and expanding the role of Roma healthcare mediators to bridge communication gaps and build trust. Policy reforms at the institutional level should promote inclusive care and address systemic barriers, while community engagement efforts can foster collaboration to design healthcare initiatives tailored to the needs of Roma women. By addressing key issues such as health literacy, communication challenges, and systemic inequities, these practical measures can reduce healthcare disparities, improve patient-provider relationships, and enhance health outcomes for Roma women. Future research should focus on evaluating the long-term impact of such interventions to ensure sustainable improvements in culturally sensitive healthcare delivery.
Reviewer 3 Report
Comments and Suggestions for Authors
Thank you very much for the article Cultural Competence of Obstetricians/Gynecologists and Midwives Providing Midwifery Care to Roma Women in Western Greece.
In general, the article has a coherent structure that aligns with the usual requirements for publication in this journal. Below, I outline some considerations that I believe could help improve your manuscript and facilitate its publication.
This is a relevant topic, given the methodology, and it addresses a common issue in healthcare services in general. However, to enhance the manuscript, it would be helpful to highlight the applicability of the study's findings and emphasize their significance.
I will provide comments section by section, following the structure of the article:
Abstract, The structure is correct and aligns with the journal’s requirements. However, I suggest adding a sentence that directly mentions the limitations or practical implications of your study's findings.
Introduction, The introduction could be improved by delving a bit deeper into how previous concepts of cultural competence have guided similar studies and how they can be more closely linked to current health policies. A new paragraph addressing this would be beneficial. Additionally, upon reviewing the bibliography you used, I suggest citing two or three more studies that discuss the impact of cultural competence. This would help provide better context for the results and discussion.
While the introduction is concise and solid, I believe it could be enhanced by expanding on the focus on the Roma population. It is necessary to justify this focus further and, as mentioned earlier, include more studies and link them to public health policies.
Methodology, To ensure better understanding, explain and clarify more thoroughly the validity of the questionnaire used and the convenience sampling technique, as these aspects are not clearly or critically addressed (sections 2.2 and 2.3). Also, review section 2.5, especially the beginning of the sentence—rather than directly explaining the type of calculation, justify its use (lines 105 to 110). Review and adjust accordingly.
Results, Please review Tables 1, 2, 5, and 6 to ensure their format is consistent. Check the information in the tables, such as adding percentages followed by the total number or including this detail in the caption. Table 2 could be improved since the information is too extensive, yet it lacks essential details. It seems that too much quantitative information has been included without aligning it with the objectives of the study—this section should be revised to make it more coherent.
Discussion, Considering the suggestions for improving the introduction, try to delve deeper into how these results could inform future interventions. While the study limitations are described, it would be helpful to emphasize their practical impact and whether the findings could be extrapolated.
It would also be important to address how these results could contribute to improving the cultural competence training of professionals and include more practical suggestions for reducing the barriers identified in accessing healthcare services.
Conclusions, Focus on improving this section, particularly by emphasizing the practical implications of the findings.
I hope these critiques help improve your article and facilitate its publication. Thank you once again for your great effort and commitment to evidence-based practice.
Author Response
(Reviewer 3)
Comments and Suggestions for Authors
Thank you very much for the article Cultural Competence of Obstetricians/Gynecologists and Midwives Providing Midwifery Care to Roma Women in Western Greece.
In general, the article has a coherent structure that aligns with the usual requirements for publication in this journal. Below, I outline some considerations that I believe could help improve your manuscript and facilitate its publication.
This is a relevant topic, given the methodology, and it addresses a common issue in healthcare services in general. However, to enhance the manuscript, it would be helpful to highlight the applicability of the study's findings and emphasize their significance.
I will provide comments section by section, following the structure of the article:
Abstract, The structure is correct and aligns with the journal’s requirements. However, I suggest adding a sentence that directly mentions the limitations or practical implications of your study's findings.
AUTHORS RESPONSE: We added this in the Abstract: The study's cross-sectional design and reliance on self-reported data may limit the generalizability and introduce bias. Enhancing cultural competence through targeted training programs can help address healthcare disparities faced by Roma women
Introduction, The introduction could be improved by delving a bit deeper into how previous concepts of cultural competence have guided similar studies and how they can be more closely linked to current health policies. A new paragraph addressing this would be beneficial. Additionally, upon reviewing the bibliography you used, I suggest citing two or three more studies that discuss the impact of cultural competence. This would help provide better context for the results and discussion.
- AUTHORS’ RESPONSE: We revised as follows: Cultural competence is increasingly recognized as a cornerstone of effective healthcare delivery, particularly in multicultural societies where patients present diverse ethnicities, languages, traditions, and health beliefs [1]. Defined as "the ability of individuals to establish effective interpersonal and working relationships that supersede cultural differences" [2], cultural competence enables healthcare providers to respect and integrate the cultural, social, and linguistic needs of patients. By addressing the multifaceted social determinants of health and fostering meaningful patient-provider relationships, cultural competence contributes to the reduction of health disparities and the improvement of patient satisfaction and equity in healthcare outcomes [2].
- The components of cultural competence include cultural awareness, cultural knowledge, cultural skills, cultural encounters, and cultural desire [3]. These elements collectively promote the ability of healthcare providers to deliver personalized, patient-centered care in diverse settings. Evidence suggests that interventions aimed at enhancing cultural competence improve healthcare providers' knowledge, attitudes, and skills, and positively influence patient satisfaction [3]. However, gaps remain in linking these outcomes to tangible improvements in patient health and adherence, highlighting the need for further research to optimize educational strategies and measure patient-centered outcomes comprehensively
While the introduction is concise and solid, I believe it could be enhanced by expanding on the focus on the Roma population. It is necessary to justify this focus further and, as mentioned earlier, include more studies and link them to public health policies.
AUTHORS’ RESPONSE: WE added this data: The Roma community, often referred to as Romani, is an ethnic minority with origins tracing back to the Indian subcontinent. Linguistic and genetic evidence suggests that they migrated to Europe over a millennium ago, during the early medieval period. Today, they are one of the largest and most dispersed ethnic minorities in Europe, with significant populations across various countries. The Roma people have a rich cultural heritage characterized by unique traditions, languages (such as Romani), and a nomadic history [4, 5].
In Greece, the Roma are an integral but marginalized part of the population. Estimates suggest that they comprise approximately 2-3% of the Greek population. While exact numbers vary due to a lack of consistent census data, it is estimated that there are about 200,000–300,000 Roma individuals in Greece.
The Roma community faces significant challenges, including profound health disparities driven by socioeconomic disadvantages, such as poverty, inadequate housing, and limited access to clean water, which contribute to higher rates of disease and poor maternal and child health outcomes. Health literacy is a critical barrier, with limited knowledge about healthcare rights and preventive services leading to underutilization and difficulty navigating healthcare systems. Cultural competence gaps among healthcare providers often result in misinterpretation of Roma cultural practices, reinforcing stereotypes and mistrust. Discrimination and stereotyping, both at an individual and institutional level, further marginalize Roma individuals, while structural barriers, such as the lack of Roma mediators and bureaucratic challenges, continue to restrict their access to equitable healthcare services [7].
Methodology, To ensure better understanding, explain and clarify more thoroughly the validity of the questionnaire used and the convenience sampling technique, as these aspects are not clearly or critically addressed (sections 2.2 and 2.3).
AUTHORS’ RESPONSE: We revised the sections as follows:
2.2. Data Collection Method
The data collection was carried out through an in-person approach at healthcare facilities, including hospitals and health centers, where midwives and obstetricians/gynecologists were employed. Researchers scheduled visits to these locations and approached eligible participants during breaks or at times pre-arranged with department heads to minimize disruption to clinical duties.
Participants were provided with a brief verbal explanation of the study's purpose and relevance, emphasizing the importance of their input in understanding cultural competence in maternity care for Roma women. To ensure participants felt comfortable, researchers explained that responses would remain anonymous, and no identifiable information would be linked to the questionnaire data.
To facilitate convenience and ensure confidentiality, participants were given the option to complete the questionnaire on-site or take it to complete later in a private setting. A secure collection method was established, where completed questionnaires could be placed in sealed envelopes or collected in a locked drop box provided at each facility. The envelopes or drop boxes were retrieved by the research team at designated intervals.
Each participant received an approximate timeframe for questionnaire completion, ensuring they did not feel rushed. Researchers remained available at the collection sites to address any logistical concerns or clarify queries without influencing the participants' responses.
2.3. Data Collection Tool
For the purposes of the study, a questionnaire was used, which focuses on the empowerment of Roma women and the fight against discrimination in health access. Permission to use the questionnaire was obtained from the researchers who developed the questionnaire. The researchers in that study investigated the healthcare access and challenges faced by Roma women in Greece, focusing on sexual and reproductive health, using mixed methods that included questionnaire that was validated for healthcare and other professionals, as well as interviews and focus groups, which revealed themes like stereotypes, cultural communication barriers, and the need for systemic changes and Roma mediators [5].
The questionnaire consists of a total of 46 items and is divided into 3 sections. The first section includes questions related to the cultural competence of health professionals regarding Roma women and consists of 20 questions in total, 4 of which concern case studies of clinical incidents and their management by the respective health professional. The second section includes questions regarding the participants' knowledge of the lifestyle and habits of Roma women and consists of 13 questions. The third section includes 13 questions for collecting demographic data. The questions are closed-ended and multiple-choice. The questions in the first and second sections use a variety of Likert scales to assess participants' responses. These include a 10-point scale ranging from "Not at all" to "Extremely" for assessing capability in providing care, and a 5-point Likert scale ranging from "Not at all" to "Absolutely", as well as "Strongly disagree" to "Strongly agree", depending on the question type. Additionally, participants were asked about their perceptions and experiences using multiple-choice questions and categorical responses, such as frequency (e.g., "Always", "Frequently", "Rarely"), and demographic questions to collect information like age, gender, education, and workplace.
Also, review section 2.5, especially the beginning of the sentence—rather than directly explaining the type of calculation, justify its use (lines 105 to 110). Review and adjust accordingly.
AUTHORS’ RESPONSE: We revised as follows:
2.5. Statistical Analysis
The choice of statistical methods was guided by the need to accurately describe the data and ensure robust comparisons aligned with the study's objectives. The Kolmogorov-Smirnov test was employed to assess the normality of continuous variables, as it is crucial to determine whether parametric or non-parametric tests should be applied, ensuring valid statistical inferences. For variables with normal distribution, means and standard deviations (SD) provided a comprehensive summary of central tendency and dispersion, suitable for identifying patterns in the data. In cases where variables deviated from normality, medians and interquartile ranges (IQR) offered a robust alternative that reduced sensitivity to outliers.
The selection of the Student’s t-test and Analysis of Variance (ANOVA) for parametric comparisons reflected the importance of leveraging their sensitivity to detect differences between groups when normality was met. Conversely, the Mann-Whitney and Kruskal-Wallis tests were preferred for non-parametric data to account for skewed distributions. The application of the Bonferroni correction for multiple comparisons demonstrated a commitment to reducing the likelihood of Type I errors, enhancing the reliability of the results.
To explore relationships between continuous variables, Spearman's correlation coefficient was utilized, chosen for its robustness against non-linear relationships and non-normally distributed data. Statistical significance was rigorously defined at a two-tailed level of 0.05, ensuring a balanced approach to hypothesis testing. All analyses were conducted using SPSS version 26.0, a standard and reliable statistical software in health research.
Results, Please review Tables 1, 2, 5, and 6 to ensure their format is consistent. Check the information in the tables, such as adding percentages followed by the total number or including this detail in the caption.
AUTHORS’ RESPONSE: We made the suggested corrections.
Table 2 could be improved since the information is too extensive, yet it lacks essential details. It seems that too much quantitative information has been included without aligning it with the objectives of the study—this section should be revised to make it more coherent.
AUTHORS’ RESPONSE: We revised Table 2.
Discussion, Considering the suggestions for improving the introduction, try to delve deeper into how these results could inform future interventions. While the study limitations are described, it would be helpful to emphasize their practical impact and whether the findings could be extrapolated.
AUTHORS’ RESPONSE: We added this: Despite these limitations, the findings provide valuable insights into the cultural competence of healthcare providers, particularly in addressing the needs of Roma women. While extrapolation should be approached cautiously, the study identifies trends and challenges that are likely applicable in similar multicultural or underserved populations, offering a foundation for future research and targeted interventions.
It would also be important to address how these results could contribute to improving the cultural competence training of professionals and include more practical suggestions for reducing the barriers identified in accessing healthcare services.
AUTHORS’ RESPONSE: We added this:
Addressing the barriers identified in accessing healthcare services, particularly for marginalized communities, requires not only understanding but also actionable strategies to improve cultural competence among healthcare professionals. For instance, integrating cultural competence training into healthcare curricula and continuing education programs can enhance the ability of providers to engage effectively with diverse populations. This training should focus on practical components such as intercultural communication, managing biases, and understanding specific community needs. For example, Roma healthcare mediators have been shown to bridge communication gaps and foster trust between healthcare providers and Roma patients, suggesting the need for similar community-specific interventions.
Further, targeted measures such as employing culturally adapted materials, providing multilingual resources, and enhancing the accessibility of services can help mitigate structural barriers. Healthcare organizations could benefit from adopting systematic policies to address discrimination and promote inclusivity. Practical suggestions, including improving health literacy through community-based workshops and leveraging technology for better service delivery, can also reduce disparities. These efforts, when combined, hold the potential to improve healthcare access and outcomes while fostering equitable and culturally sensitive care.
Conclusions, Focus on improving this section, particularly by emphasizing the practical implications of the findings.
AUTHORS’ RESPONSE: We revised the Conclusions section as follows: The findings of this study highlight the critical role of cultural competence—encompassing knowledge, skills, attitudes, and awareness—in improving healthcare delivery for Roma women in Western Greece. Healthcare providers with greater knowledge of Roma cultural, social, and healthcare challenges exhibited higher levels of cultural competence, emphasizing the need for practical, targeted interventions. These include integrating structured cultural competence training into medical, nursing, and midwifery curricula, implementing continuous professional development programs such as workshops and community immersion activities, and expanding the role of Roma healthcare mediators to bridge communication gaps and build trust. Policy reforms at the institutional level should promote inclusive care and address systemic barriers, while community engagement efforts can foster collaboration to design healthcare initiatives tailored to the needs of Roma women. By addressing key issues such as health literacy, communication challenges, and systemic inequities, these practical measures can reduce healthcare disparities, improve patient-provider relationships, and enhance health outcomes for Roma women. Future research should focus on evaluating the long-term impact of such interventions to ensure sustainable improvements in culturally sensitive healthcare delivery.
I hope these critiques help improve your article and facilitate its publication. Thank you once again for your great effort and commitment to evidence-based practice.
Reviewer 4 Report
Comments and Suggestions for Authors
Keywords: Roma; obstetricians/gynecologists…. We list after a comma
Line 68; The sample for the study was collected through convenience sampling. Please explain what this means.
Line 78: Please add the name of the questionnaire and who the author is.
Line 126: elated roles – proszę o wyjaśnienie
As for future considerations, it is necessary to consider conducting a study on a much larger study group.
Nevertheless, the article is very interesting, carefully presented, well thought out, and the conclusions are very important.
Author Response
Author's Reply to the Review Report (Reviewer 4)
Comments and Suggestions for Authors
Keywords: Roma; obstetricians/gynecologists…. We list after a comma
AUTHORS’ RESPONSE: We made the correction.
Line 68; The sample for the study was collected through convenience sampling. Please explain what this means.
AUTHORS’ RESPONSE: We added this: This non-probability sampling approach selected individuals who were readily accessible and willing to participate, making it a practical choice for exploratory research. While convenience sampling facilitates data collection in specific contexts, it may limit the generalizability of findings due to potential selection bias.
Line 78: Please add the name of the questionnaire and who the author is.
- AUTHORS RESPONSE: The questionnaire is provided in Supplementary File 1 and the reference is Asimaki, E.; Vivilaki, V.; Valasaki, M. A Mixed Methods Study on Challenges for Roma Women and Access to Health in Greece. J. Midwifery 2023, 7(Suppl. 1). https://doi.org/10.18332/ejm/172138.
Line 126: elated roles – proszę o wyjaśnienie
AUTHORS’ RESPONSE: We made the correction.
As for future considerations, it is necessary to consider conducting a study on a much larger study group.
AUTHORS’ RESPONSE: We added this: Crucially, studies should aim to include larger and more diverse sample groups to enhance the generalizability of findings and better represent the broader population.
Nevertheless, the article is very interesting, carefully presented, well thought out, and the conclusions are very important.
Round 2
Reviewer 1 Report
Comments and Suggestions for Authors
We thank the authors for addressing all suggested changes. No additional changes are needed.
Author Response
Reviewer 1
We thank the authors for addressing all suggested changes. No additional changes are needed.
AUTHORS’ RESPONSE: We would like to express our deepest appreciation for your insightful comments and for accepting our revised manuscript. Your feedback has been invaluable in enhancing the quality and clarity of our research.
Best regards
Reviewer 2 Report
Comments and Suggestions for Authors
Thank you for making corrections, they are satisfactory. I accept the article for printing.
Author Response
Reviewer 2
Thank you for making corrections, they are satisfactory. I accept the article for printing.
AUTHORS’ RESPONSE: We would like to express our deepest appreciation for your insightful comments and for accepting our revised manuscript. Your feedback has been invaluable in enhancing the quality and clarity of our research.
Best regards
Reviewer 3 Report
Comments and Suggestions for Authors
I would like to express my gratitude for the revisions and suggestions made to the manuscript. I deeply appreciate the effort and recommendations, which have significantly improved its quality.
However, I would like to provide some additional recommendations that you may consider:
Table Format and Consistency:
Table 1 has readability issues that could be improved.
Table 2 might benefit from a reduction in size and a review of its formatting.
Tables 5 and 6 present noticeable differences in their design. I recommend unifying the formatting criteria for all tables, in line with the journal’s requirements.
Bibliography in the Introduction:
Adding bibliographic references to lines 69-72 would strengthen the arguments presented in this section.
Similarly, incorporating additional references in lines 48-55, where some concepts are repeated, could enrich the content. This is, however, an optional suggestion.
I am confident that these minor adjustments will further enhance the manuscript’s presentation and align it with the journal’s standards.
Best regards
Author Response
Reviewer 3
I would like to express my gratitude for the revisions and suggestions made to the manuscript. I deeply appreciate the effort and recommendations, which have significantly improved its quality.
However, I would like to provide some additional recommendations that you may consider:
Table Format and Consistency:
Table 1 has readability issues that could be improved.
AUTHORS’ RESPONSE: We revised Table 1 as you suggested.
Table 2 might benefit from a reduction in size and a review of its formatting.
AUTHORS’ RESPONSE: We revised Table 2 as you suggested.
Tables 5 and 6 present noticeable differences in their design. I recommend unifying the formatting criteria for all tables, in line with the journal’s requirements.
AUTHORS’ RESPONSE: We revised the Table 6 as you suggested.
Bibliography in the Introduction:
Adding bibliographic references to lines 69-72 would strengthen the arguments presented in this section.
AUTHORS’ S RESPONSE: We added References 7 and 8.
Similarly, incorporating additional references in lines 48-55, where some concepts are repeated, could enrich the content. This is, however, an optional suggestion.
AUTHORS’ S RESPONSE: We added References 3 and 4.
I am confident that these minor adjustments will further enhance the manuscript’s presentation and align it with the journal’s standards.
Best regards

Reviewer 4 Report
Comments and Suggestions for Authors
Very good article. Congratulations to the whole team!
Author Response
Reviewer 4
Very good article. Congratulations to the whole team!
AUTHORS’ RESPONSE: We would like to express our deepest appreciation for your insightful comments and for accepting our revised manuscript. Your feedback has been invaluable in enhancing the quality and clarity of our research.
Best regards